# Social Media's Role in the Changing Religious Landscape of Contemporary Bangkok

**Ruchi Agarwal \*** and **William J. Jones**

Social Science Division, Mahidol University International College, Nakhon Pathom 73170, Thailand;
william.jon@mahidol.edu
**\*** Correspondence: ruchi.aga@mahidol.ac.th

**Abstract:** In this article, the authors argue that social media is becoming a more influential medium for religious services in Bangkok, Thailand. Buddhism, with an attendant mixing of Thai animistic beliefs in spirits, inanimate objects, and talismans, along with other expressive forms of religious enterprise, is adapting to a changing personalized media landscape to reach ever more diverse audiences. Social media in particular is allowing for social and personal entrepreneurship of religious expressions that formerly were largely limited by physical space and its attendant costs. The authors argue that utilizing the Uses and Gratification approach will provide a powerful method for understanding and evaluating how and why religious entrepreneurship is shifting in contemporary Thailand.

**Keywords:** Ganesh; social media and religion; Uses and Gratification; religion and media; religious entrepreneurship





## 1. Introduction

The impacts of technology in the last 30 years on the lives and lived experiences of people have had dramatic effects on how people engage with one another on interpersonal and societal levels. Social media has dramatically transformed social spaces to both enlarge the public square of social engagement and compress time for interaction between strangers and fellow travelers. Interconnections through the fast uptake of internet connectivity and dramatic expansion of smartphone usage have combined with enlarging effects of social media to create novel 'sacral spaces' for entrepreneurs and consumers of religious tokens, events, and ceremonies. These digital 'sacral spaces' offer entrepreneurs and consumers new opportunities to engage with one another on physical and virtual levels. Furthermore, this allows the aforementioned to find new pathways and opportunities to achieve and gain gratification through religious ceremonies and associated religious paraphernalia that enrich people on a material and spiritual basis.

The authors argue that social media has allowed for the creation of novel sacred spaces which religious entrepreneurs can take advantage of by finding new markets and consumers, expanding their customer base, and lowering overhead costs. Religious consumers can likewise take advantage of these sacred spaces to engage in personal spiritual gratification. The authors will use the framework of Uses and Gratification Theory (UGT) in order to analyze the behaviors and intent of participants in religious activities in Bangkok, Thailand. The paper will first provide a literature review and place the study within the prevailing literature. Second, it will discuss how internet connectivity and social media use have evolved in Thailand. Third, it will detail the methodology of UGT and how it is a strong explanatory tool for understanding religious behavior in this study. Last, it will analyze primary data collected from interviews with religious entrepreneurs and consumers in Bangkok with discussion and findings from interviews and how UGT explains intent of action and analyzes personal behavior around new sacral spaces.

## 2. Religion and Social Media

The literature covering the effects that media, specifically social media, has when interfacing with religion is quite expansive. That said, there are a few important spheres within the literature that are important to highlight and demonstrate how this study fills a gap within the current literature. From a macro perspective that addresses how religion and social media interact on a broad social plane, Hoover (2010) explores how social media has altered the ability of individuals to represent themselves with regard to their individual religious interpretations and navigate with personal clarity their specific 'brands' or religious interests. The notion of the 'self' is key to how social media allows for a personalized interface with religion directly and how this is selectively received by consumers. There is a broad literature concerning how social media has impacted religious practices. Knott et al. (2013) explore how media has amplifying effects on secular feelings in Britain, finding that media often allows for specific discursive narratives to be built to exclude certain religious views in secular and diverse societies. Öncü (2006) finds similar conclusions in Turkey within the context of media's narrative construction of mediating mass religiosity and trying to craft narratives in line with state ideology of a secular Muslim state. Gillespie and Nauta (2013) examine the effects of social media and extremists and martyrdom content on the rise and the solidification of Islamophobia. Similarly, McVicar (2013) explores how media has effectively provided a platform for narratives of religious extremism put to dominion theology in the United States. This is similar to Kgatle (2018), who finds social media, specifically Facebook, having a strong impact on the rise and establishment of prophetic churches in Africa. Coman and Coman (2017) argue that in post-secular societies, Facebook can act as a vector for religious figures to exercise influence during times of crisis. Cheong et al. have studied how social media has allowed for the enlargement of space for religious figures and organizations to expand their influence as well as methods of communication to amplify inaction online (Cheong and Poon 2009; Cheong et al. 2009, 2013). Brummans et al. (2013) and Taylor and Van Every (2011) find that religious authorities in digital spaces have a dynamic interaction within a negotiated order of asymmetry that allows for a space of increased choice of engagement.

The important takeaway from these studies is how social media allows for novel space to be created as a mediator between individuals which did not exist previously. Secondly, the studies are largely either macro in nature, addressing social-level phenomena, or asymmetrical from authority figures to followers. Previous studies do not approach the twin issues of religion and social media from a perspective of markets of production and consumption. The issue of authority and social media amplification is not novel, but the intent of entrepreneurs and consumers is so far understudied. This exploratory study of religious entrepreneurs and consumers attempts to fill a gap in the existing research surrounding personal motivations, how social media enhances their need for specific gratifications within the religious sphere, and how social media allows for Gratifications Sought and Obtained.

## 3. Economic Growth and the New Media in Thailand

Thailand, a predominantly Theravada Buddhist country, experienced strong economic growth in the late 1980s and the early 1990s. Between 1987 and 1996, the economy grew by an average of 9% (Asian Development Bank 2015). Economic growth gave rise to a new middle class with enhanced resources, which made them prime consumers in the new media market. Changes in consumption patterns were evident in the rise of cinemas equipped with new sound technology and increased satellite and cable television subscriptions.

The new mass media played a major role in further developments of the Thai religious marketplace. Newspapers increased sales by covering sensational stories involving occult practices. In 1997, Thai newspapers and magazines displayed images of former prime minister Chavalit Yongchaiyut's wife carrying a good luck elephant statue and dancing to fulfill a vow to the spirits to further her husband's political career (Kitiarsa 2002, pp. 166–67). Television additionally promoted spirit mediums, fortune tellers, and magic monks, popularizing their activities to a wider audience. Even monks opposed to such practices utilized media; when respected Phra Phayom publicly criticized the practice of channeling spirits, he inadvertently increased public awareness of mediums and popular religious practices.

The government also liberalized media (Siriyuvasak 2000, pp. 99–113), giving rise to cross-border information flows via modern telecommunication technology, transportation, mass media, and the internet. Lewis (2005) refers to Thailand as 'Virtual Thailand.' He sees the boundaries between representation and reality as ambiguous in Thai culture. Thailand's aim to be a trade and tourism center has pushed the country increasingly towards a digital economy. Lewis (2005) notes that Thailand's geographical position, relatively large population, openness to foreigners, and economic success have allowed it to remake itself as a trade and tourism center (Lewis 2005, pp. 1–3). The Thai government engaged in building the digital infrastructure with new major telecom networks built in the early 1990s with an addition of three communication satellites launched. The prioritized IT sector gave rise to new IT entrepreneurs, including former prime minister Thaksin Shinawatra, Boonchai Bencharongkul, and Sonthi Limthongkul. Later, the head of the agribusiness Charoen Pokphand (CP) also became involved in the telecom business by winning a landline phone contract in 1991 (ibid. 9–10). The country later adopted the internet as a commercial tool, with its first commercial internet service provider (ISP) starting in 1995. The Internet Thailand Service Centre (later known as Internet Thailand) was the first with a commercial ISP license granted by the Communications Authority of Thailand (CAT) in 1994. It began its operation in 1995, followed by the KSC Comnet. There were 2.3 million internet users by 2004, accounting for 3.8% of the Thai population (Liu 2014, pp. 76–77). Since 2004, over 100 internet service providers have been established, with the top two operated by Chulalongkorn University and King Mongkut Institute of Technology.

## 4. Internet Connectivity and Social Media

Internet infrastructure was improved with the launch of 3G in 2013 and gave a significant boost to online shopping and online marketing in Thailand's retail market. Social media sites such as Facebook, Twitter, and YouTube emerged as necessary advertising instruments for retailers. The latest numbers from May 2017 published in an article in the Bangkok Post show that Thailand ranks in the top 10 for social media usage with an average increase of 20% annually, raising the number of users to 47 million on Facebook, 11 million on Instagram, and 9 million on Twitter. The latest statistics reveal that Bangkok remains the world's biggest city, with 30 million Facebook users in January 2017, most of the users logging in through their cellular devices. A Bangkok-based social media analytics developer sees social media as increasingly becoming an essential tool for business marketing and business success (Fredrickson 2017, May 24).

According to Datareportal (Kemp 2021), with a population of 69.71 million, Thailand's Internet penetration rate is 75% of the population, with 52 million internet users in January 2020. The most active internet users remain in urban centers such as Bangkok, Chiang Mai, and Phuket. A rapid increase in internet users has given rise to social media, and sites such as Facebook have become increasingly important. Furthermore, Thais spent an enormous amount of time online or using the internet at 9 h and 1 min per day and using social media for 2 h and 55 min of those 9 h (Kemp 2021).

Thailand is also the second-largest market for cellular devices or smartphones in Southeast Asia after Indonesia. The country has experienced fast growth in social media usage due to the increase in cellular market penetration. With a large rural population, internet penetration has remained low but has been counterbalanced by impressive growth in the mobile penetration rate. In 2015 alone, 23 million smartphone devices were sold in Thailand (The Nation Thailand 2021). Smartphone usage is at 52.71 million persons out of nearly 70 million (Statista Research Department 2021a). According to WeAreSocial, Hoot suite (January 2020), there are now *93.39 million mobile connections* and *42 million mobile social media users* in Thailand (Kemp 2021). In 2017, there were an estimated 47 million Facebook users in Thailand (9th in the world), with Bangkok being the top city with 27 million Facebook users alone (Fredrickson 2017). Social media users in general in 2020 numbered 52.71 million (Statista Research Department 2021b). Facebook is the most trafficked network within social media platforms, catching 52.37 million users (Statista Research Department 2021c). Facebook and YouTube ranked second and third behind Google among websites' monthly traffic, with Facebook at 465,800,000 with an average time per visit of 13 min and YouTube with an average of 461,900,000 and an average time per visit of 29 min 15 s (Kemp 2021). These numbers indicate that Facebook and YouTube far outstripped the Thai language portal of Pantip.com in fourth with 123,300,000 monthly traffic and 4 min 44 s average visit time (Ibid.).

The above data demonstrate two essential characteristics of uses gratification. Mobile device penetration and usage are very high in Thailand, and social media usage is also predictably high for the Thai population. The next variable is social media engagement or sharing, which brings social media usage and its ability to bridge individuals and find connections for religious expression. Facebook statistics regarding connectivity and share reach among its Thai market is interesting in that it reported 79% of FB users to have advertisement reach (Ibid.). Facebook further reports user reach in the following dimensions (see Tables 1 and 2).

**Table 1.** Thai Social Media Engagement Percentages.

| Platform | Reach Matrix | Percentage |
|---|---|---|
| Facebook | Average Post Reach vs. Page Likes | 8.1% |
| Facebook | Average Organic Reach vs. Page Likes | 5.2% |
| Facebook | Percentage of Pages Paid Media | 23.4% |
| Facebook | Average Paid Reach vs. Total Reach | 32.7% |
| Facebook | Average Engagement Rate for any FB Post | 3.05% |
| Facebook | Average Engagement Rate for FB Video Post | 9.52% |
| Facebook | Average Engagement Rate for FB Image Post | 4.15% |
| Facebook | Average Engagement Rate for FB Page Link Post | 3.16% |
| Facebook | Average Engagement Rate for FB Page Status Post | 2.51% |

Adapted by authors from Datareportal Data.

**Table 2.** Thai Social Media Usage.

| Data Platform | Data Matrix | Individual Usage |
|---|---|---|
| Internet | Time Spent Online | 9 h 1 min |
| Social Media | Time (Day) Online | 2 h 55 min |
| Facebook | Monthly Traffic/Time Visit | 465,800,000/13 min |
| YouTube | Monthly Traffic | 461,900,000/29 min 15 s |

Adapted by authors from Datareportal and Statista.

The above table demonstrates that Thai internet and social media users are not just active in the general regard. Rather, they are involved in sharing and engaging within the social media network to an extraordinarily high degree.

Furthermore, Bangkok is the city with the highest number of active Facebook users worldwide. Thai's 9 h of online connectedness is also ranked third in social media usage in the world, and lastly and most importantly, Thailand has the world's top Facebook engagement at 6.99% (David 2018). These numbers above are aligned with the theoretical framework that the authors will utilize to demonstrate a connection between religious gratifications and social media usage that was previously referred to at length.

## 5. Uses and Gratification and Social Media: A Methodology for Examination

Uses and Gratification Theory argues that media usage has a purpose and that consumers of religion utilize media formats and platforms to fulfill desires and/or needs that are intrinsic to the consumer on an individual and social basis (Blumler and Katz 1974; Lin 1996). This initial working context is somewhat vague and requires specificity. Rubin et al. (1994) provides needed distinction in differentiating between Gratifications Sought (GS) and Gratifications Obtained (GO). Furthermore, they argue that Gratifications Sought are those that people select by focusing on the content of the medium in question. In contrast, Gratifications Obtained are those material and/or metaphysical outcomes that an individual receives or attains.

Media platform usage is central in determining both how and why religious identification and religious commodification occur among religious suppliers and consumers in Bangkok. Developed by Rosen et al. (2013), the media and technology usage scale was previously used to measure users' technology and media usage. This 60+ measuring scale includes video gaming, internet search, and mobile phone usage indices. While this scale is useful in measuring broad technology usage, it does not necessarily benefit small-scale case studies or focus group phenomena as it is far too broad in scope. However, connections between mobile phone usage, social media usage, and interconnected social media platforms provide a more focused research model. Ratcliff et al. (2017) find interesting and salient research results central to the article, namely correlations between uses and gratifications and social media usage. The authors find a strong statistical correlation between social media usage and an individual's religious leanings that fulfills individually desired needs. The authors found, using regression analysis, that predictive variables of smartphone usage, general social media usage, media sharing, and social media friendships were strongly correlated to personal needs of the time, self and learning. Media sharing (beta = 0.237, $p < 0.001$) and general social media usage were predictors of personal religious needs related to time (beta = 0.37, $p < 0.001$) and smartphone usage (beta = 0.17, $p > 0.05$) contributing strongly to needs related to self (Ratcliff et al. 2017, p. 19). Time, self, and learning in this study specifically relate to an individual's feelings towards media and its ability to serve religious needs relating to time spent, self-gratification, and ability to learn and share one's religious beliefs.

This methodology is the framework in which this article attempts to analyze individuals' usage of social media, the purposes for which they use social media, and its ability to provide gratifications dependent on the usage of social media networks. Uses and Gratification allows for a three-point direct correlation between:

1. The effects of social media on religious space;
2. Religious entrepreneurs' and consumers' individually understood gratification needs;
3. How social media is used in order to obtain their needs.

Lastly, UGT—in conjunction with primary data collection using the qualitative research method of in-depth interviews and simple purposeful and snowball sampling—allows for a richer understanding of personal intent. The authors will demonstrate plausible connections between individual wishes and the intent of individuals with regard to their specific behaviors concerning religious entrepreneurship and consumption, and whether or not they achieved their stated intentioned goals. The application of UGT to personal accounts of religious behavior will address gaps in the current literature.

## 6. Social Media as a Medium for Religious Engagement

Historically, traditional media platforms have been the primary medium to disseminate information for spirit mediums, monks, and those with religious authority in place of physical, interpersonal interactions. However, with the advent of social media, religious figures can now personalize their services directly to consumers with minimal cost and bypass traditional entry barriers. Increasing trends towards digital religiosity could threaten the existence of (self-defined) religious experts, thus forcing them to adapt to the rapid changes. As a diasporic Hindu Brahmin mentioned during an interview:

> I increasingly maintain my presence on social media sites like Facebook and Instagram. It is an important platform to reach a bigger audience and advertise the religious services I provide. It costs nothing, and that's where I get noticed.

(Pandit G. 2016)[1]

Pandit G. maintains his presence on Facebook with regular updates about his services (see Figure 1); he regularly updates his Facebook page with the different religious events he has been part of. Some of his spiritual services, as seen in Figure 1, include baby blessing ceremonies, the installation of a new Ganesa image ritual, the *plook sek*[2] ritual for an icon seller shop in a shopping mall, and conducting a blessing ceremony for a new beauty clinic that opened in Bangkok. Regular updates as such allow Pandit G. to maintain his virtual presence and also help to advertise the variety of religious services he can offer. His presence on Facebook has allowed him to widen his reach to the spiritual service seeker by providing a virtual platform where he can easily and freely promote his religious services. He explains further:

> People don't know what kind of ceremonies we as Brahmins can do. I post photos of the various religious ceremonies I perform, and many people regularly follow my updates. Sometimes my regular followers share my posts with their friends looking for such spiritual service providers. People are always interested in conducting different religious rituals, but they don't know how to do it and need someone who can perform the ritual in the right way. Facebook has become a more accessible medium for connecting the seeker and provider. Earlier, people used to go to temples to seek such service providers but were still reluctant to ask. This may be because of the language barrier, as well as sometimes people are not sure if the Brahmin from the temple would go to perform rituals outside of the temple. Facebook gives an open space to post real pictures, and I don't need to explain much as the actual images explain things by themselves.

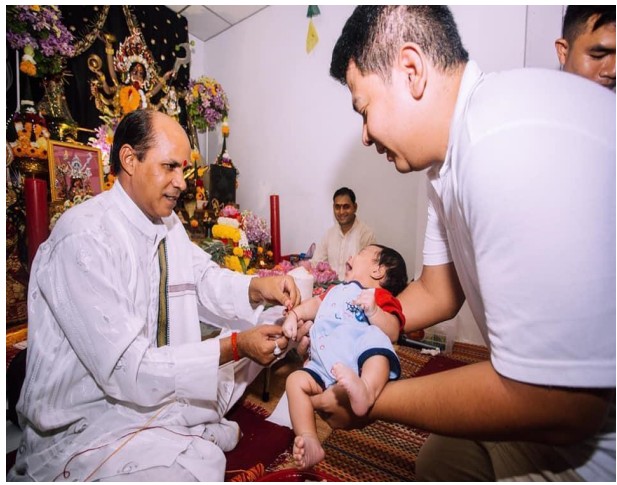
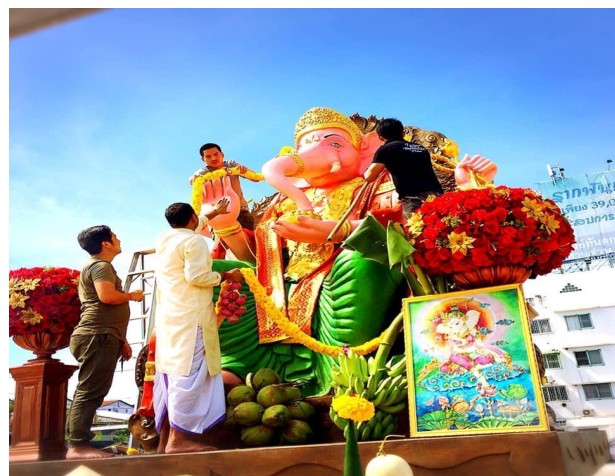
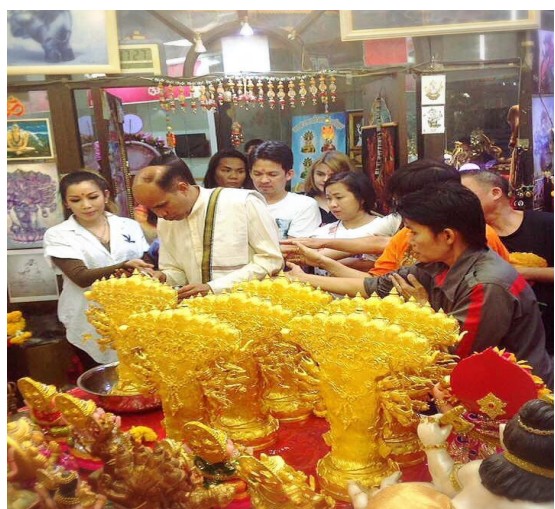
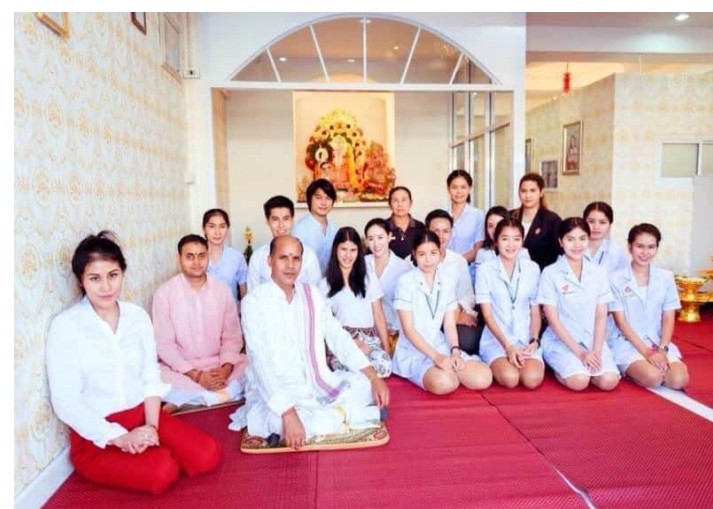

**Figure 1.** Images show, left–right, top–bottom: Blessing newborn baby, posted on 22 January 2019. Installing a new Ganesa image in Rangsit area, posted on 13 February 2019. Conducting the ceremony for a shop selling religious icons in Imperial Mall in Ladprao area, posted on 15 February 2019. With devotees at a beauty clinic after performing pooja at their clinic in Bangkok, posted on 27 February 2019 (Images used with permission).

Many diasporic Brahmins provide religious services independently or through Hindu temples in Thailand. Many do not speak fluent Thai, which becomes a communication barrier. However, many Brahmins have learned the Thai language and have experienced Thai ways of conducting rituals.

Facebook has resolved the problem of reaching out to the Thai-speaking Brahmin. Religious service seekers can easily reach out to the right person and see the reviews of the service provided through pictures and comments that come along with the Facebook posts. Below in Figure 2 are Facebook posts of the diasporic Brahmin (Mr. Y), a musician and a singer invited to many religious events organized by diasporic and non-diasporic communities in Thailand. Mr. Y has a group, including other diasporic Brahmins, who perform the religious services, while Mr. Y[3] himself provides the musical component to the religious services. After each event, Mr. Y updates his Facebook page with photos from the events, which allows him to showcase the religious services he provides and use these posts as online promotional material. His posts receive many likes, and some of his followers even share the posts further on their timelines.

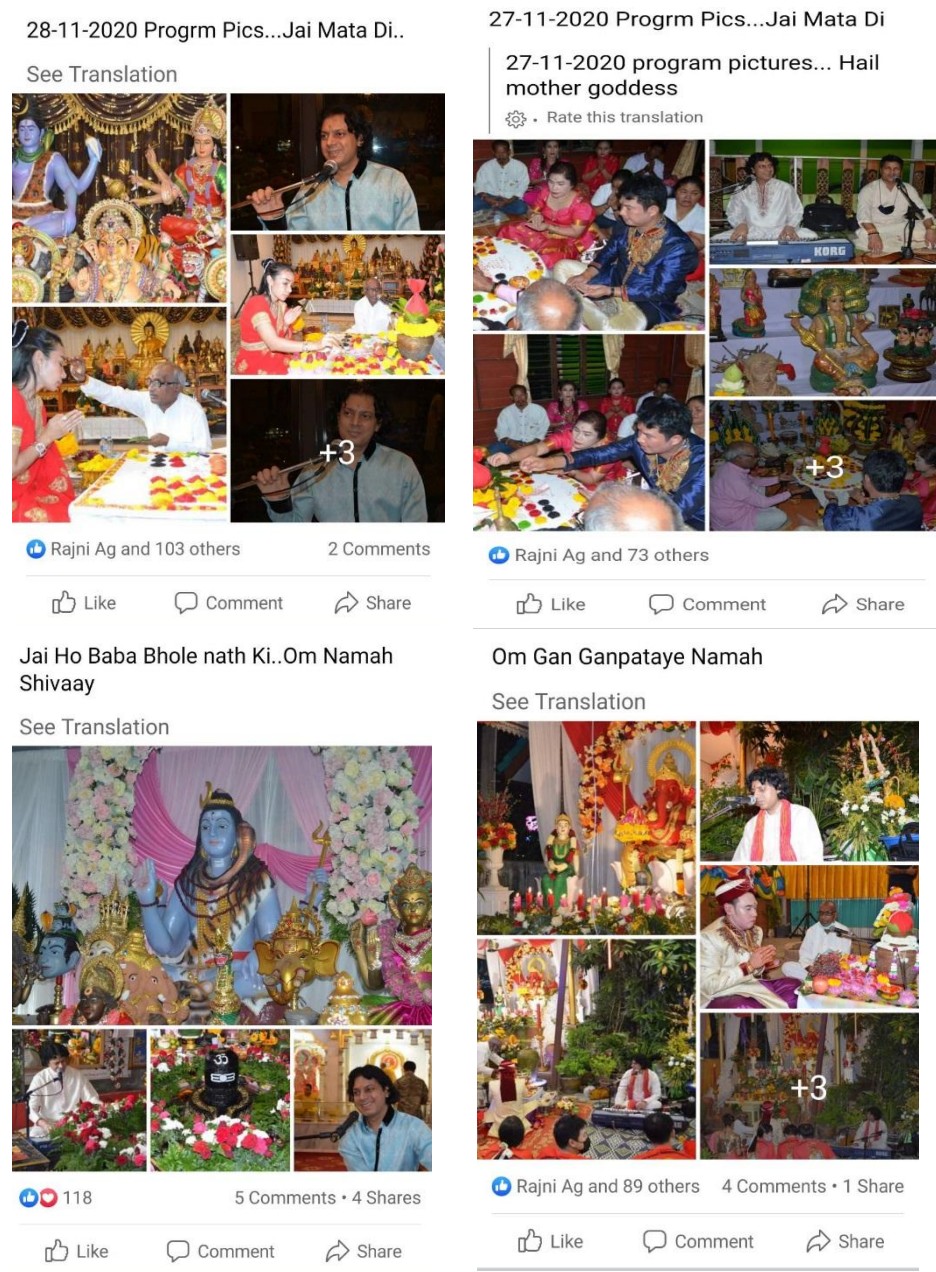

**Figure 2.** Mr. Y shares pictures from different events he has been invited to by both Indian and Thai devotees. Source: Mr. Y's Facebook page. Images used with permission.

From the perspective of Gratifications Sought, one can see the variety of uses for social media originating in monetary gain to solidify traditional religious ritual rites. For example, Pandit G. and Mr. Y were able to sustain and even grow their business interests by hosting ceremonies and religious events. Social media is the primary mechanism used for Gratifications Obtained as this allows for virtual networking and no-cost marketing.

Pandit G. needed to enlarge his following and open digital space for more prospective followers within the traditional diasporic community of Brahmin worshippers. However, years of engagement have allowed Pandit G. to not only solidify the traditional market, but also expand into the hybrid religious hybrid market of Thai worshippers who are outside of the Indian diasporic community. Furthermore, the ability to repost old photos from previous events on Facebook proves to be the driving factor as the repost feature helped the providers sustain a living during the COVID-19 pandemic. During the pandemic, significant religious events and gatherings have more or less been either banned or postponed

due to the fears of the further spread of the virus and risk-averse nature of an unvaccinated population unwilling to physically meet. This unintended consequence can be considered an additional value creation in Pandit G.'s original material intent. Mr. Y is in line with Pandit G., who was able to enlarge his audience for material benefit, which was the original intent of his Gratification Sought.

### 7. Media and the Popularization of Thai Religiosity

The case of Thailand is particularly interesting to further understand how the media plays a role in popularizing religiosity. Thailand is a predominantly Theravada Buddhist country; however, there is evidence of a strong mix of Hinduism (see Coedes 1975; Hall 1981), Chinese traditions, as well as animism and spirit mediumship. The result is a hybrid form of religiosity where one can find a Thai worshipper paying respect to Buddha at a Thai temple, making offerings to Ganesa (see Agarwal and Jones 2018) during the Ganesa festival, seeking the advice of a spirit medium to correct problems in their day-to-day life, as well as burning the paper offerings at the San Chao during Chinese New Year festival. This is just one of the many examples of hybrid religiosity evident in the Thai context. Thai religiosity is therefore unique, with Thai Buddhists modifying their beliefs by adopting useful religious practices that protect them in their daily lives.

As anthropologist Hendry (1999, p. 12) has explained, religions are often utilized as a shock absorber by individuals faced with external stresses and strains wrought by economic dislocation, technological development, and social competition. Religious traditions can serve multiple functions, providing psychological and social support networks. In the globalized world, new media has allowed individuals to explore and pick and choose aspects of different religions that fulfill their desires. It is important to note that the new media provides a means through which broadcasts or features can be directly promoted, disseminated, expressed, and revitalized to a wider range of audiences (Lindgren 2022, pp. 3–5). Thus, a platform is available for religious seekers and providers to meet and become involved in religious exchanges. These exchanges have been increasingly evident since the economic boom in Thailand. Assessing the boom's impact on urban Thais, Phongpaichit and Baker (1998) note how an upwardly mobile Thai middle class sought to improve their lives through education, hard work, and modifying traditional merit-making practices. Instead of passively supporting monks to gradually accumulate merit and improve spiritual standing in a future incarnation, the new urban population brought religious practices in line with their immediate aspirations. This as well included seeking advice from religious experts such as priests, monks, fortune tellers, and spirit mediums.

These religious experts provided not only personal advice and healing, but also help with finding missing objects and winning lotteries. As Keng, part-time fortune teller, explains:

> . . . my customers are looking for anything that provides mental and physical comfort to their busy and uncomfortable lives. They come with their problems and ask for possible solutions to improve their present and future. I direct them to make merits at Buddhist temples, worshiping a Hindu god, offer donations to schools, old age homes, hospitals, etc., or sometimes even spirit mediums depending on the individual problems of my clients. I visit spirit shrines and also seek advice from my teacher, who taught me about astrology.
>
> (Keng 2015)[4]

Concern over the future of business fortunes was a boon for the fortune-telling business. Prediction and intervention services flourished, as did spirit mediums. From the 1960s, their popularity steadily increased (Kitiarsa 2002, pp. 166–67; Kitiarsa 2012, pp. 16–17). Urban spirit cults incorporated Chinese and Hindu beliefs leading to the growth of the amulets and talisman industry. Magic monks helped in reading omens, telling fortunes, protecting people from evil spirits, and providing help/advice in acquiring wealth (lottery numbers, business fortunes) (ibid. 39–40). Politicians, military leaders, and high-ranking government officials have also sought blessings and guidance from popular spirit mediums and monks.

By the mid-1990s, Thais were spending around THB 20 billion annually on spirit–medium services (Kitiarsa 2008, p. 138).

The growing awareness, popularity, and expansion of hybrid religiosity in Thailand result from the new media. Here, as elsewhere, new technologies have given rise to stepped-up flows of information, reworking the relationship between religion, popular culture, and politics. These changes have resulted in the creation of novel public spaces for religious worship, a process that Herbert (2011, p. 627) termed 'publicization'. Herbert (2011) notes how publicization has increased the visibility of religious symbols and discourses, making them available for mobilization, contestation, and criticism by religious and non-religious actors. An example of publicization is of Mr. P below, who found the solution to his problems with the help of new media. As Mr. P explains:

> I was looking for a solution to the financial problems in my business and sought the help of a spirit medium . . . he advised that I sought the help of Ganesa and asked me to join Ganesa worship. I searched online for Ganesa–related religious events. I found a Facebook post promoting a big Ganesa festival organized at a Thai temple by a group of people on the outskirts of Bangkok. I came to the temple after a long bus ride changing buses at least three times. I was so determined to join the festival that I spent over two hours traveling. I had no previous knowledge of Ganesa, and I did not know how to worship him. So I sought the help of other participants in the festivals to know how I was supposed to worship Ganesa to resolve my financial problems. I thank FB for helping me find what I was looking for. Had there been no FB, it would have been challenging for me to find quick relief to my problems. Joining the festival today is mentally soothing for me, and for the rest, Ganesa will help me.
>
> (Mr. P. 2015)[5]

Social media provided a mixed outlet for problem-solving in a monetary sense and compensation for fear of bad luck befalling people. The example of Keng is quite instructive of the multiplier effect that social media has to provide consultation and advice to people seeking solace and peace of mind in the consumer sphere of religious protection. These examples of Gratifications Sought are assistance in times of distress and superstitious fear of bad luck befalling people. They were able to Obtain Gratification through the medium of social media.

The cases of Keng and Mr. P are two sides of the same phenomenon—the use of religion for personal compensatory goals for those facing adversity. In the case of Keng, Gratifications Sought were to expand on an existing consumer base for those in need of spiritual guidance. Mr. P was the seeker of spiritual guidance during a period of adversity and sought psychological reinforcement. In both cases, Gratifications Obtained are in accordance with Gratifications Sought—material compensation and religious protection and guidance.

## 8. Online Faith and A New Sacral Space

Internet marketing is an important part of the process of publicization. Online retailing has experienced substantial growth in Thailand over several years, creating a virtual marketplace for retailers and consumers. Increased use of the internet in everyday life also means an increasing number of digital sites to access religious information. Personal interviews with Brahmins, spirit mediums, fortune tellers, and icon sellers have revealed that online presence is essential to disseminate religious information, initiate learning, and survival. Social media is the new medium of social communication, previously dominated by physical spaces, which allows for a new sacral space where spirituality can flourish.

Fortune tellers increasingly use social sites such as Facebook to maintain their virtual presence and promote themselves among spiritual consumers. One of the progressively used features of Facebook is streaming video. Figure 3 shows three examples from 20 February 2019 when three different fortune tellers provided live consultation for fortune seekers. Witchy Tarot Reader, the woman in the first picture, provided her services during the

afternoon when fortune seekers are likely to have some free time during their lunch break. She charges THB 99 per question and requires her customers to pay via wire transfer during the live consultation. The man in the second picture above gave free consultations for two questions and was reading after 6 p.m., which is appealing to the after-office work market. The third picture is a Facebook page that posted a picture of a personal shrine instead of a particular fortune teller. The description above the picture notes that in addition to fortune-telling, they also conduct rituals to correct misfortunes for THB 200 per consultation. Each entrepreneur had different approaches to reaching their customers, targeting other groups at other times of the day and providing differentiated services such as tarot card reading, fortune-telling based on birth charts, and even specialized rituals to cure misfortune. Thus, Facebook has provided a new sacral space for providers and seekers of spiritual help.

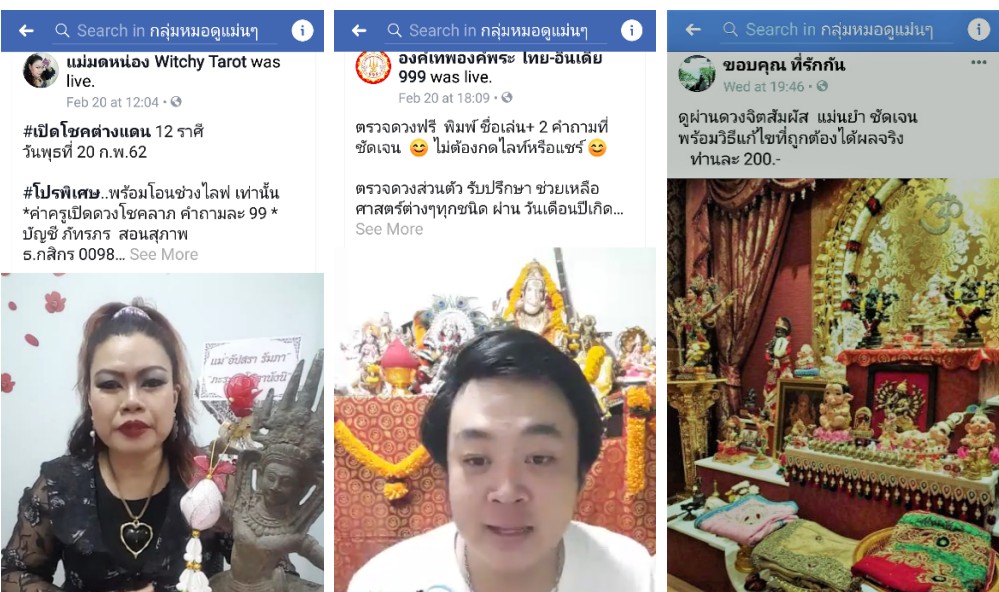

**Figure 3.** Tarot card reader and Fortune Teller providing services on social media site Facebook. The Thai text refers to the different promotions the fortune tellers have for the visitors of the FB page. For example, in the first image the tarot reader will provide her services to only those who pay for her services during the live broadcast on 20 February 2019. The fortune teller in the second image answers two questions of those who watch his live broadcast for free. He also provides personal fortune reading and consultation to those interested. The third image mentions that their fortune reading is very accurate and gives quick results and a fee of 200 baht is charged for the services. Source: Accurate Fortune Tellers' Group, Facebook. Retrieved on 20 February 2019. Images used with permission.

In the past, access to sacral spaces was dependent on the availability of transportation and information. Social media has made information available within seconds and has done away with the physical limitations of transport. One can search for physical sacral places, directions to reach them, histories of the particular sacral site, and at the same time, receive quick reviews from earlier visitors. Places with more positive reviews tend to attract an increasing number of visitors.

This is not to say that temples or shrines without a virtual presence lose their importance. Spiritual consumers still visit temples and shrines to respect deities and religious figures and make merits. The importance of social media is that it has made spiritual information highly accessible. Social media has created a new virtual sacral space where people can search for the information they need, check other users' products/services, access real-time feedback, and receive updates on the latest happenings in the religious marketplace.

An example of this is a dhamma retreat center which established its Facebook fan page in 2015. The physical location of this retreat center is in the north of Thailand. Within a short period, their fan page had over 3000 followers. The admin of their fan page, Ms. N, mentioned:

> The owner of our dhamma retreat center wanted to open this Facebook fan page because it's easy to sell merits. We have only 100 statues for people to buy at 5999 baht. Part of the money we get from selling these statues through our Facebook page was used to build a Rahu temple. I feel that Facebook allows us to reach out to more people. We can boost the page by paying a small fee and reach an even wider audience.
>
> (Ms. N, 2016)[6]

New sacral spaces have seen an expansion of spiritual advertising on social media. Spiritual advertising traditionally was by word-of-mouth along with print media. However, word-of-mouth publicity has gained additional amplification through social media sites such as Facebook. Social media benefits both the seeker and the provider at the same time. The seeker can easily search and find, whilst the provider's maintenance of virtual presence on social media is highlighted with word-of-mouth publicity through sharing of Facebook posts and personal reviews posted by previous consumers. Mrs. K mentioned during an interview:

> I just bought land to start a new business, and I want to make sure that the spirits guarding the land are happy. I was unsure which spiritual master to invite to perform the ceremony to please the guardian spirits. My friends recommended I seek the help of Acharn T to perform the land blessing rites. I didn't know Acharn T, but I checked his Facebook profile and found numerous posts of him performing rituals. Seeing these posts, I was convinced that Acharn T was the right person to invite, and I am happy with my choice.
>
> (Mrs. K. 2016)[7]

Another respondent, Ms. Pac, a Ganesa devotee, mentioned her regular visits to a Ganesa temple annually to attend the Ganesa festival in Nakon Nayok province. In an interview with her at a temple in Bangkok, Ms. Pac further mentions:

> I have never come to this temple before as I usually visit Uthayan Ganesa in Nakon Nayok. I buy a little Ganesa statue at the Uthayan premises to float in the river after the ceremony. But this year, I chose to come to Phrom Rangsi temple. I never knew that this temple organizes the Ganesa Festival. I saw the event on Facebook the organizer posted. One of my friends shared this post with me and immediately added the organizer to my Facebook friend list to get additional details. The organizer was extremely friendly and answered all my questions through the chatbox. Had I not seen the Facebook post, I would go to Nakon Nayok as usual. But here, it's so convenient, the temple is within Bangkok, and I can save time and travel costs.
>
> (Ms. Pac. 2015)[8]

An interview with Ms. O revealed that she had been following a Facebook page owned by a writer who owns a publishing house and writes books on Hindu beliefs and practices in the Thai language. Since the books are in the Thai language and use simple language, she became a fan follower of this writer. By accident, one day, she found him on Facebook and started regularly following updates posted by this writer. She found out that the writer also organized religious ceremonies at major Hindu festivals and invited Hindu priests to conduct the rituals. On the occasion of Janam Ashtami,[9] the writer organized a ceremony and posted an open invitation for anyone to join. Seeing this post, Ms. O, a big fan of this writer, decided to join the event. This allowed her to see the writer in person and accumulate merits by being part of the religious ceremony. Ms. O expressed her excitement several times and shared a picture of the Ganesa statue (see Figure 4). She also received a

prize from the writer at the ceremony. She says that she strongly connects with this Ganesa statue and pays respect regularly by offering flowers and fruits. She stated that had she not found her favorite writer through Facebook, she would not have had the opportunity to meet the writer and receive a spiritual gift that holds utmost significance to her.

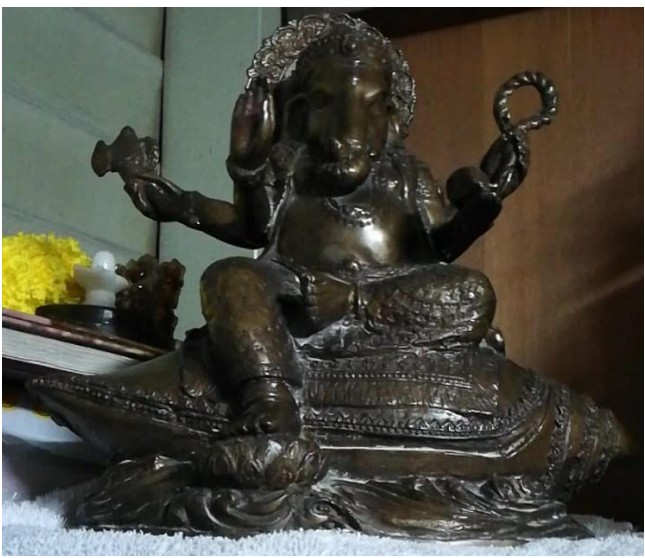

**Figure 4.** A personal statute of Ms. O that she received as a prize at the Janam Ashtami event. Photo credit: Ms. O (2015). Images used with permission.

Another popular page on Facebook is the GaneshLuckshmee, started by Mr. N, who later planned to open a physical shop in Future Mall Rangsit. Mr. N mentions:

> I bought a Ganesa statue for myself from a Thai sculptor but did not do any rituals. I had wished for something, and I got it. Later a friend of mine visited my house and saw my Ganesa and liked it. I start to post on Facebook, and people begin to share these posts of mine.
>
> (Mr. N, 2016)[10]

Mr. N stated that he did not intend to open any online page or physical shop initially but started with a Facebook page for himself when a friend of his posted a photo of Mr. N's personal Ganesa statue. Several people admired this random post. Seeing people's interest in this Thai-crafted Ganesa statue, Mr. N started to add new posts on Facebook more often. When he received more queries via Facebook, he decided to open an online shop to promote Thai craftsmanship. He further decided to open a physical shop (as seen in Figure 5) in a shopping mall close by as the number of customers from the online shop increased. Mr. N further explained:

> I didn't intend to open the physical shop, but I did because of the increasing customer demand. I've opened a Facebook page for one year, and the customer's request for visiting or praying at the shop. Having both bits of help, because, for example, the customers first order for the 4000-baht statue, but if they visit the shop and see the bigger and more beautiful one, they upgrade the size. I feel online, and physical shops complement each other.
>
> (Mr. N, 2016)

Asked how he promotes his shop and artistic and spiritual work, Mr. N said:

> Just calling my friends to help clicking like. People keep sharing. Advertise through Facebook two times. But the customer will come more from word of mouth. There are some customers that win the lottery by worshipping the statue from my shop. But I'm not daring to post it on Facebook. I think it's more like you are making a viral. I want people to have faith in themselves. I don't promote

that much but will post like a game for people to play and give away things (see Figure 6). I might post more since I just quit my job to do this.

(Mr. N, 2016)

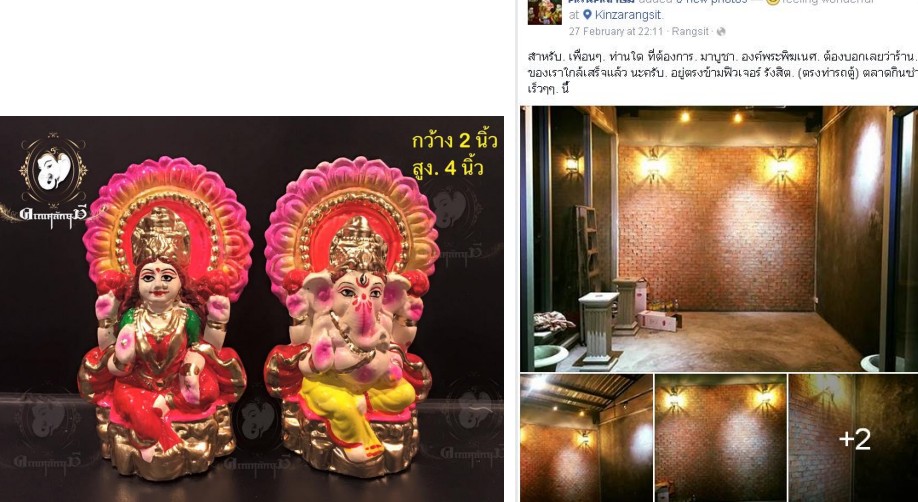

**Figure 5.** A pair of Laxmi and Ganesa (2-inch-wide and 4-inch-tall) was posted on GaneshLuckshamee's Facebook page on 20 January 2016; a new physical shop in Future Mall, Rangsit, under construction by GaneshLuckshamee, was posted on 27 February 2016. The Thai post invites visitors to the physical shop located in Rangsit area, currently under construction. Images used with permission.

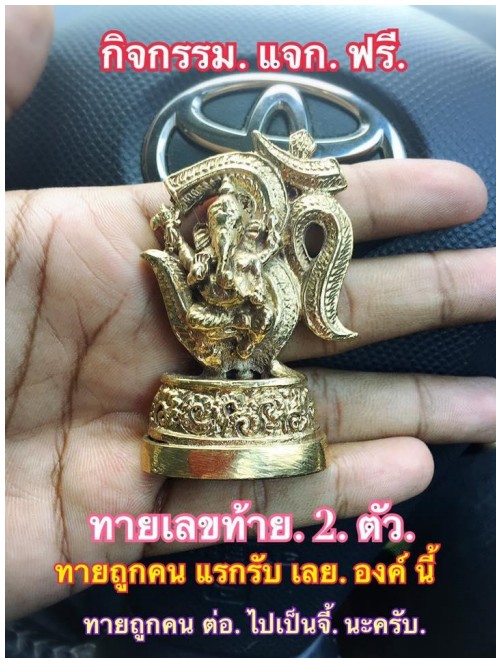

**Figure 6.** This is an example of a game 'free give away to those you can predict the last two digits of winning lottery numbers' post shared by Mr. N on his Facebook page. Photo credit: Mr. N. Images used with permission.

The example of Mrs. K and Ms. Pac demonstrates social media's effect on advanced networking and increased information flows. For Gratification Sought, Mrs. K and Ms. O participated in religious events for spiritual nourishment and enjoyment. Mrs. K was able to Obtain Gratification through social media, offering an opportunity to view and, through direct communication, find new sacral spaces to practice her religious faith. Ms. O was

able to network and find new religious teachers to expand her religious knowledge and engagement with more people in different religious settings. Ms. Pac was able Obtain Gratification through the accumulation of religious icons that are personally highly coveted. She shows these off to her friends and strangers and shares her story of luck attached to religion with people at religious events. The Gratification Sought is hard to define in this case as it was through luck she was able to obtain the Ganesa idol, but it can be inferred that there was previous knowledge of the idol and the wish to someday be able own one. Mrs. K sought good fortune by utilizing a Buddhist monk in order to sanctify her place of business in order to increase the probability of good luck. Mr. N's intent was materialist in nature to expand his business and income opportunities without the associated capital investment in a physical shop. The Gratification Obtained is in line with the Gratification Sought—increasing the entrepreneur's ability to generate income.

## 9. Conclusions

It should be noted that Gratifications Sought across all interviews are two-fold—monetary and spiritual. On the monetary side, it has been shown that social media allows for an expansion of space and expansion of the market. The ability to attract consumers outside of traditional spaces and networks with physical limitations has enabled religious entrepreneurs to tap into new markets and engage in real-time spiritual business without the monetary overhead of traditional practices. Secondly, those who are spiritual consumers are able to come across new networks and experiences that were previously not possible by the opening of virtual marketplaces that allow for novel connections. Gratifications Obtained by both entrepreneurs and consumers ranged from monetary to spiritual. Entrepreneurs were able to expand their consumer base, enjoy greater flexibility in providing religious goods and services, market their services to a broader consumer base and lower overhead capital costs. Additionally, virtual social spaces allowed for pandemic compensation as an entrepreneur was able to derive income by reposting old events whilst conducting spiritual services in the midst of COVID-19 and its limitations on physical contact. Spiritual consumers were able to expand their channels of religious communication and consumption by virtually trying religious services that were heretofore limited to word-of-mouth and physical presence. Furthermore, they were empowered to find, collect and obtain idols and tokens of religious significance by being exposed to religious services and rituals that were unknown to them previously. The ability to connect to religious entrepreneurs that were unknown and revered demonstrates the power of social media to provide direct connections that were not before possible. The primary data collected in this study demonstrate that the Uses and Gratification Theory is a strong analytical lens to understand the increasingly diverse environment of religious engagement in Thailand's religious consumer market. Social media has offered an avenue for people to gain monetarily, network and expand connections with sacred sites, offer information for protection and problem-solving, and provide joy and happiness through luck and connection to religious idols.

The authors have argued that the Uses and Gratification Theory provides a compelling means of analysis for understanding the shifting digital space, opening avenues for advanced participation and intent of participants. The conclusions drawn can be expanded upon with further research into the effects that social media, internet connectivity, and mobile devices have on religious consumers and entrepreneurs' ability to network, lower costs, and expand engagement for many personal reasons. This paper has demonstrated that new media platforms enabled by the internet have provided a new virtual space for religious entrepreneurs and consumers to further expand their ability to satisfy religious needs through Uses and Gratification Theory.

**Author Contributions:** Investigation, R.A.; Writing—review & editing, W.J.J. All authors have read and agreed to the published version of the manuscript.

**Funding:** This research received no external funding.

**Institutional Review Board Statement:** The study was conducted in accordance with the Declaration of Helsinki, and approved by the Institutional Review Board (or Ethics Committee) of Mahidol University Social Science IRB (protocol code 2016/006.1901 and 19 January 2015) for studies involving humans.

**Informed Consent Statement:** Informed consent was obtained from all subjects involved in the study.

**Data Availability Statement:** Not applicable.

**Conflicts of Interest:** The authors declare no conflict of interest. The research for this project was self-funded by the researchers.

## Notes

[1]  Pandit G. is a diasporic Hindu Brahmin. He provides freelance religious services to diasporic Hindus in Thailand as well as the local Thai devotees. He was interviewed on 1 May 2016 at the Hindu Samaj Temple.

[2]  A Thai ritual performed by monks or priests to anoint religious icons or amulets.

[3]  Mr. Y maintains a virtual presence through Facebook. The researcher has been following his page for over 3 years and has used his posts with permission.

[4]  Keng is a fortune teller in Bangkok, and she was interviewed on 20 December 2015.

[5]  Mr. P is a retired pilot who now runs his own business and was interviewed during a Ganesa festival held in a Thai temple on the outskirts of Bangkok in September 2015.

[6]  Ms. N is a 26-year-old female who works as an administrator of the Facebook page of a dhamma retreat center located in Chiang Mai. She was interviewed on 5 March 2016 at the dhamma retreat center in Chiang Mai.

[7]  Mrs. K, a businesswoman in her early 40s, was interviewed at the diasporic Hindu temple, Dev Mandir aka Hindu Samaj, on 2 April 2016.

[8]  Ms. Pac is a common laborer in her early 20s. She was interviewed at a Buddhist temple, Phrom Rangsi Temple, on 20 September 2015.

[9]  A Hindu festival that celebrates the birth of a Hindu god, Krishna, on the eighth day of the dark fortnight of the month of Bhadrapada. The festival falls in the month of August–September annually.

[10] Mr. N, aged 28, is the owner of the GaneshaLuckshamee Facebook page. He was interviewed on 29 February 2016 at his shop in Future Mall, Rangsit.

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
