# Peer review of "Social Media’s Role in the Changing Religious Landscape of Contemporary Bangkok"

_religions, doi:10.3390/rel13050421_

Round 1

Reviewer 1 Report

This article has a clear problem formulation,  the theoretical approach is well presented, as are the empirical examples.  The aarticle brings new insights into the field of popular religiosity.

Author Response

Thank you for the review and comments 

Reviewer 2 Report

Uses and Gratification and Social media’s Role in the Changing Religious Landscape of Contemporary Bangkok presents some interesting materials and suggestions which could add to our collective understanding of the mechanics of contemporary religiosity and the role of social media in them. That said, I feel the manuscript needs some more attention to realize its potential.

General comments

  1. The article needs thorough language editing. Some sentences are almost impossible to understand due to grammar mistakes and missing or superfluous words (e.g. p.1, par 2, line 1 “media medium”; line 2 “…how and why religiously identified and religious commodification…”). I will not list all such mistakes I found (also as I am not a native speaker of English myself), but this needs attention.
  2. The title is confusing. I would either insert “Theory” after Gratification, or just keep “Social Media’s Role in the Changing Religious Landscape of Contemporary Bangkok”. I prefer the second, as it highlights what the article is actually about, rather than the theoretical framework it uses to interpret its findings.
  3. I suggest reconsidering the structure of the article to facilitate the readers’ understanding.
  • First present the link between economic growth and the rise in media use. I suggest also including a short paragraph on religion in Thailand/Bangkok in general, as scholars reading this from e.g. a media-studies background will not necessarily know much about this.
  • Next, indicate how both the economic growth and the rise in social media use have contributed to a shift in religious practices, towards ‘prosperity religions’. It may be interesting to elaborate on that term a bit.
  • Provide the theoretical and methodological set-up of the article. This would include the presentation of the Uses and Gratification Theory that opens the manuscript as it is now (although I would omit the statistics, see comment below), but also the way the authors have studied this. From the text, I understand that interviews were conducted, and social media were analyzed, but I would appreciate a more explicit discussion of method, e.g. How were respondents selected?. This section should end with a line like ‘In the next section, a number of cases will be presented which illustrate different uses of social media for religious purposes.
  • Presentation of cases
  • Reiteration of how the cases presented fit into the Uses and Gratifications framework
  • Reiteration why this is relevant and indicative of ongoing changes in religious praxis in Thailand
  1. Delete unnecessary repetitions to streamline overly long passages (cfr. specific comments).
  2. Capitalization of title and section-headers (and function titles in the main text) is not consistent.
  3. Unsure if the journal requires this, but the Facebook pages from which the screenshots presented in the article are taken are not included in the References section.
  4. A number of the cases presented look at the figure and role of religious/spiritual entrepreneurs in the digital realm. If the authors would want to elaborate on this, I can think of two good entry-points:
    1. Campbell & Lövheim. "Introduction: Rethinking the online–offline connection in the study of religion online." In Information, Communication & Society8 (2011): 1083-1096 briefly discusses puja-offering websites.
    2. Andrea Jain’s 2014 book. Selling Yoga: From Counterculture to Pop Culture can offer some ideas on the processes of commodification and branding that are part of the publicization and popularization the article talks about.

Specific comments

Introduction, Uses and Gratification and Social Media

Page 1, Par. 1:   The definition of GS is not understandable.

Page 2, Par 1:    The statistical data presented seem at odds with the more ethnographical rest of the article. The factors ‘time, self, learning’ are also not entirely clear. I suggest to rewrite this paragraph in a more descriptive style (i.e. without the statistics, but focusing on the findings and how they are relevant for the discussion presented in the article), or omitted altogether if not relevant (enough).

Economic Growth and the New Media in Thailand

Internet, Social Media, and Internet Connectivity

Page 2, par 3:    I feel this section could be significantly shorter. Details such as the names of early IT-entrepreneurs and companies are not so very relevant to the discussion at hand.

Page 3, par.3:    Again, this section contains some unnecessary repetition.

Page 4, par 1:    The use of ‘#1’ is unconventional in academic writing. I suggest you rewrite these sentences, e.g. Bangkok is the city with the highest number of active Facebook users worldwide.

Page 4, par 4:    The excerpt from Fredrickson 2017 is so self-evident I would omit it altogether, or just incorporate this fact into the text with a reference.

Social Media as a Medium for Religious Engagement

Page 6, par 1:    The repeated use of ‘a number of’ in this paragraph draws attention to the fact that no exact data are presented. If the authors have collected data that enable them to give an exact or approximate percentage/number they should do so. If not, consider re-writing to omit at least some of the ‘a number of’s’.

Media and the popularization of Thai religiosity

Page 7, par 2:    To say that the economic boom of the 80s and early 90s coincides with the rise of new media is slightly problematic. Although this period saw much of the development of new media technologies, they really only became available to the broader public from the early to mid 90s onwards…

Page 7, par 3:    The word ‘cult’ has some negative connotations due to which some publishers avoid its use entirely. Although I do not advocate banning the word, I would suggest considering alternatives.

Publicization of Thai religiosity

Online faith and A New Sacral Space

Page 9, par 1:    “The internet has become an integral part of the daily lives of the Thai population.” Undoubtedly true, but why is it stated here. Perhaps more at home in the intro and/or conclusion.

Page 9, par 2:    “…and also for survival.” I assume this is meant to denote economic/financial success?

Page 10, par 1:  “…with just the click of a mouse.” Earlier in the article, the authors emphasized the share of mobile devices in Thai’s use of the Internet and digital resources. These don’t usually have a mouse.

Page 10, par 2:  ‘Sacral space’. Is this the same as sacred space? Existing research distinguishes between online spaces where religious actions are performed (potentially ‘sacred’ space), and online pages where information on religion is to be found. Perhaps it would be beneficial to also make this distinction here, as there is a qualitative difference between noting religious entrepreneurs have moved online, and saying religious practice has moved online. Potentially interesting sources on digital religion not included in the references section that discuss digital sacred spaces include the works of Heidi Heidi Campbell and Chris Helland.

Page 10, par 3:  The excerpt from the interview is difficult to understand. Is it possible that the ‘tell’ should be ‘sell’? If not, I really don’t understand and the excerpt requires more explanation.

Page 11, par 1:  In the excerpt, we read ‘One of my friend shared’ (instead of one of my friends). If the interviewee made this mistake, consider adding ‘(sic.).’ If not, correct the mistake.

Page 11, par 2:  Why is the name of the writer not mentioned?

Discussion and Findings

The structure of this section is very confusing. Structuring words such as ‘Secondly’, and ‘Next’ seem to have been used arbitrarily.

Author Response

We would like to thank the reviewer for the very detailed comments which helped us to come up with a much better quality of our manuscript. 

The manuscript has been thoroughly edited according to the comments and suggestions including the English editing by the native speaker. 

As suggested the title has been adjusted by not including the term Uses and Gratification. 

The entire manuscript has been restructured to give a smoother read to our paper. We have incorporated all the comments suggested. 

Reviewer 3 Report

Brief summary: the paper argues that the Uses and Gratification Sought and Obtained approach powerfully explains how Thai religious users are using social media to reach new customers and combine online and offline spaces for commerce.

The various sections need reorganization, because the article starts with methodology rather than an introduction to religion and how users use social media. The paper should start more broadly and then narrow down to the question of the research, for example, with discussion of how religious people in Thailand engage in religious practices and then perhaps how the internet has spread and finally combining the two trends in religious social media usage by religious people. As it is, the sections don't smoothly connect.

Also, you assert claims without evidence, such as that Thailand is the second largest market for cell phones after Indonesia (can this be true? China is third?). In any case, you lack a citation or source for the claim, and similarly for others, such as the 9% economic growth rate. In the section on mobile phones and Facebook, you have much extra information. Only give the information necessary to make your case! If it doesn't relate directly to your topic, then cut it out. If it is important, then link it directly to Thai religious services. There is a mass of detail that is unnecessary (or unconnected).

By page 7, the paper continues to add new ideas and there is no clear logic or aim that I can see. Where is the theoretical framework of Gratifications Sought and Obtained on pp. 8-10? This is the body of the paper but it is purely descriptive and therefore not linked to these ideas.

I suggest you take the several findings in the conclusion and rework them as section headings: social media opens up new religious spaces, the flow of information and visual representations opens access to new religious authorities, and so on. Then, in each section, the theoretical (or what you've called methodological) approach of GS and GO should be used to illustrate differences and how it helps to see something new. In addition, it isn't clear what scholarship you are extending or critiquing or adding to.

Your paper lacks a relevant literature background, a clear focus, and linkages between the theory and your work. In addition, most of the paper is simply descriptions, without any connections to your theoretical discussion.

Author Response

First of all, many thanks to the reviewer for giving very useful comments and suggestions. The manuscript has been restructured extensively according to the comments provided. 

The Gratification model has been explained in detail as suggested as well as the literature review, discussion, and results presented in the revised manuscript are much more extensive than the previous version.

Round 2

Reviewer 3 Report

This is a much improved paper. Congratulations! 

I have a few suggestions: first, please re-read carefully (possibly out loud) because many sentences are sentence fragments that don't have a complete subject and verb.

Second, I would suggest that you make Mrs./Ms. K's quote a block quote on page 12, as the other quotes are. Also, in some places, you refer to "Ms. K" and in other places it is "Mrs. K."

Third, in your conclusion, you helpfully summarize Gratifications Sought and provide two examples (monetary and market). It would also help to summarize Gratifications Obtained and illustrate with examples in your last two paragraphs as well.

Fourth, please add a reference for the economic growth rate of 9%.

Author Response

Thank you again for the very useful comments.

We have got the entire manuscript proofread by a native speaker to ensure there are no grammatical errors. 

Second, I would suggest that you make Mrs./Ms. K's quote a block quote on page 12, as the other quotes are. Also, in some places, you refer to "Ms. K" and in other places it is "Mrs. K."

We have already corrected the mistake in the revised manuscript.

Third, in your conclusion, you helpfully summarize Gratifications Sought and provide two examples (monetary and market). It would also help to summarize Gratifications Obtained and illustrate with examples in your last two paragraphs as well.

The conclusion section has also been revised as suggested by the reviewer.

Fourth, please add a reference for the economic growth rate of 9%.

We have already added the reference as well.